# Electrochemical Properties of Super Austenite Stainless Steel with Temperature in a Green Death Solution

**Hyun-Kyu Hwang [1] and Seong-Jong Kim [2,***

1. Graduate School, Mokpo National Maritime University, Haeyangdaehak-ro,
   Mokpo-si 58628, Jeollanam-do, Republic of Korea
2. Division of Marine Engineering, Mokpo National Maritime University, Haeyangdaehak-ro,
   Mokpo-si 58628, Jeollanam-do, Republic of Korea
* Correspondence: ksj@mmu.ac.kr

**Abstract:** In this investigation, potentiodynamic polarization experiments were conducted on UNS S31603 and UNS N08367 in a modified green death solution, which simulates the environment of a desulfurization device (scrubber), using temperature as a variable. A Tafel analysis showed that the corrosion current density of UNS S31603 at the highest temperature (90 °C) was approximately 4.5 times higher than that of UNS N83067. A surface analysis using a scanning electron microscope revealed that pitting and intergranular corrosion occurred simultaneously in UNS S31603, whereas UNS N83067 exhibited a stronger tendency toward intergranular corrosion. After electrochemical experiments, the corrosion rates according to maximum damage depth were compared with the corrosion rates according to corrosion current density; the relationships between the two values were expressed as α values. The α values of UNS N08367 were higher than those of UNS S31603, indicating that the local damage rate of UNS N08367 was higher.

**Keywords:** UNS S31603; UNS N08367; modified green death solution; temperature; potentiodynamic polarization

## 1. Introduction

$NO_x$ and $SO_x$ contained in harmful exhaust gases generated by ships are major air pollutants. Since 2015, the International Maritime Organization (IMO) has tightened regulations on ships operating in emission control areas (ECAs) [1]. Ships are obligated to use fuels with a sulfur content of no more than 0.1%. In 2020, IMO regulations were expanded to all seas except for ECAs [1]. Ships are obligated to use fuels with a sulfur content of no more than 0.5% or to install exhaust gas aftertreatment devices using SOx/NOx reduction technologies that produce equivalent or better results. Since the use of fuels with a low sulfur content places an economic burden on companies, the use of exhaust gas aftertreatment devices is spreading. The exhaust gas aftertreatment device was used for early power plants and was made of carbon steel [2].

However, the device faced corrosion problems due to the operating environment, including high temperatures and washing water [3]. Experts have devised surface treatments, corrosion inhibitors, and material developments as corrosion prevention measures for exhaust gas aftertreatment devices at power plants [4]. However, the surface treatment and plating are difficult to apply to the field because the internal shape of the exhaust gas aftertreatment device has many welds and bends. When applying this technology, the non-uniform surface treatment reduces the lifespan and exposes the base material quickly. For this reason, it is judged that the development of corrosion inhibitors and materials to solve the corrosion problem faced by exhaust gas aftertreatment devices is important. On the other hand, Phull B. S. et al. investigated the corrosion prevention of carbon steel, stainless steel, and titanium by applying a corrosion inhibitor after manufacturing a laboratory scale exhaust gas aftertreatment device [5]. As a result of the research, great damage was

observed to the carbon steel due to a high corrosion rate and, in the case of stainless steel and titanium, pitting damage was suppressed but crevice corrosion still occurred [5]. In addition, corrosion inhibitors have recently been developed with environmentally friendly organic materials [4–6], but exhaust gas aftertreatment devices for ships have open-loop systems due to the use of seawater and due to internal environments with high temperature and acidity, so there is essentially no application research.

Therefore, various stainless steels with reduced manufacturing costs have been developed thanks to technological improvements [7]. Austenitic stainless steel, which has excellent corrosion resistance, wear resistance, and heat resistance, is generally used as a material for exhaust gas aftertreatment devices on ships. However, these devices for ships use seawater for washing water, accelerating local corrosion damage due to sulfate and chloric acid generated during the desulfurization process [8]. In addition, because exhaust gas is exposed to a harsh, corrosive, high-temperature environment, the corrosion problem still occurs in an exhaust gas aftertreatment device made of general austenitic stainless steel. Accordingly, super austenitic stainless steel with improved corrosion resistance and pitting resistance, achieved by increasing the content of chromium, molybdenum, and nitrogen among the stainless steel components was applied to the exhaust gas aftertreatment device material for ships.

Therefore, many investigations have been conducted on the green death solution, simulating the desulfurization device environment, for super-austenitic stainless steel and general austenitic stainless steel [9–12]. Cardoso et al. analyzed the pitting resistance of UNS S31603 and UNS N08367 sensitized by heat treatment at a temperature of 25 °C in a green death solution [10]. Rhodes investigated the corrosion resistance of nitridized UNS S31603 in a green death solution [11]. Baek et al. conducted experiments on the pitting resistance of welded UNS N08367 in a green death solution [12]. As such, corrosion behavior and pitting resistance analyses of UNS S31603 and UNS N08367 have been conducted, but analyses of the temperature variables for UNS S31603 and UNS N08367 in a green death solution are rare. Temperature affects corrosion behaviors and rates. As the temperature increases, the movement of electric charges increases, affecting the lifespan of the materials. Since exhaust gas aftertreatment devices are subjected to wide temperature ranges, investigation on the effects of temperature is necessary. Therefore, in this study, electrochemical experiments on UNS S31603 and UNS N08367 were conducted in a modified green death solution, using temperature as a variable, and the degrees of damage at temperature were analyzed and compared.

## 2. Experimental Method

Electrochemical tests were conducted with general austenitic stainless steel UNS S31603 and super-austenitic stainless steel UNS N08367 in modified green death solution using temperatures as a variable.

Table 1 shows the chemical compositions of UNS S31603 and UNS N08367, which were selected based on the pitting resistance equivalent number (PREN). The PREN was calculated as follows [13]:

$$PREN = \%Cr + 3.3 \times \%Mo + 16 \times \%N \tag{1}$$

The PRENs of UNS S31603 and UNS N08367 were 23.6 and 45.6, respectively. Thus, the PREN of UNS N08367 was about twice that of UNS S31603. Specimens were selected for comparisons between the corrosion resistance and the PREN values. To minimize thermal deformation, the specimens were cut into a size of 2 cm × 2 cm using a fine-cutter supplied with cooling water and then polished using increasingly finer sandpaper (up to #600). To remove foreign particles generated during processing, ultrasonic cleaning was performed for 1 min, followed by washing with acetone and distilled water. The processed specimens were inserted into holders designed for electrochemical experiments, with only an effective area of 1 cm$^2$ exposed during the experiments. The components of the electrochemical cells consisted of a specimen used as a working electrode, a platinum counter electrode,

and a silver/silver chloride (Ag/AgCl) reference electrode, which was used for potential monitoring during the experiments [14].

**Table 1.** Chemical compositions of UNS S31603 and UNS N08367 (wt. %).

| | Ni | Cr | Mo | C | Si | Mn | P | S | Cu | N | Fe |
|---|---|---|---|---|---|---|---|---|---|---|---|
| **UNS S31603** | 10.19 | 16.7 | 2.03 | 0.0232 | 0.60 | 1.05 | 0.034 | 0.0028 | 0.282 | 0.012 | Bal. |
| **UNS N08367** | 24.62 | 20.62 | 6.44 | 0.015 | 0.27 | 0.72 | 0.017 | 0.001 | 0.53 | 0.232 | Bal. |

Table 2 presents the chemical composition of the modified green death solution, which contains sulfuric acid ($H_2SO_4$) and hydrochloric acid (HCl) and simulates the environment of a desulfurization device [15]. The temperatures were selected based on the flow characteristics with the internal structure of the wet scrubber [16]. This study reported that the temperature of the exhaust gas approached 100 °C due to the gas's concentration phenomenon and that after the gas passed a certain height, a uniform temperature of no more than 30 °C on average was maintained because heat exchange with the cleaning solution hardly occurred.

**Table 2.** Chemical composition of modified green death solution.

| Chemical Composition | Concentration |
|---|---|
| Sulfuric acid ($H_2SO_4$) | 16.9 vol. % |
| Hydrochloric acid (HCl) | 0.35 vol. % |
| Distilled water | Bal. (total: 1000 mL) |

Based on these findings, the experiments in this study were conducted at temperatures of 30 °C, 60 °C, and 90 °C. In the electrochemical experiments, active dissolution reactions were generated by artificially raising the potential of the working electrode. A potentiodynamic polarization experiment was conducted to identify the corrosion characteristics and corrosion resistance of the material within a short time [17]. After immersion for 1800 s as the initial delay time before the experiment, the potential was applied at a scan rate of 1 mV/s from –0.25 V to 1.2 V based on the open circuit potential. In the potentiodynamic polarization experiment, reproducibility was confirmed after three to five experiments, and the corrosion current density was calculated through the Tafel extrapolation method after the experiment. The Tafel extrapolation method was automatically calculated with EC-Lab's software in the range of −0.25 to +0.25 (vs. Ag/AgCl) based on the open circuit potential. After the electrochemical experiment, the damage ratio was calculated using the Image J software and the surface was observed using a 3D microscope and a scanning electron microscope (SEM). The ratios of surface damage and maximum damage depth were calculated as average values after nine measurements and after excluding the largest and smallest values.

## 3. Results and Discussion

Figure 1a,b depicts the potentiodynamic polarization curves of UNS S31603 and UNS N08926 in the modified green death solution. The current density tended to increase when the OCP shifted to the negative direction in the cathode polarization curve. It can be assumed that an activation polarization reaction occurred as hydrogen was generated ($H_2O + 2e^- \rightarrow H_2 + 2OH^-$) with the exchange of electric charges at the interface between the metal surface and the solution [18]. UNS S31603 and UNS N08367 exhibited different passivity characteristics. UNS N08367 found the passive state characteristics of stainless steel in an environment containing chloride ions ($Cl^-$), whereas UNS S31603 exhibited the passivity characteristics of stainless steel in an environment containing sulfate ions ($HSO_2^{2-}$) [18,19]. An active dissolution reaction ($Fe \rightarrow Fe^{2+} + 2e^-$) was observed in UNS

N08367 [18], with the current density steadily increasing as the potential rose. On the other hand, UNS S31603 reached a critical current density as the potential increased and a clear passive current density was observed thereafter [19]. The corrosion potential of UNS S31603 had negative values, whereas that of UNS N08367 had positive values at all temperatures. As the nitrogen (N) content in stainless steel increases, the potential becomes nobler because N is more inactive than other elements [20]. Therefore, a high corrosion potential was observed in UNS N08367, which affected the occurrence of the pitting potential. The pitting potential of UNS N08367 was approximately 0.95 V. On the other hand, no pitting potential was observed in UNS S31603 in the given polarization range. It can be assumed that UNS N08367 and UNS S31603 had different OCP values. The corrosion potential of UNS N08367 was nobler than that of UNS S31603. As a result, in the electrochemical test condition (0–1.2 V vs. OCP), the potential of UNS N08367 increased to more than 1.2 V. However, since UNS S31603 has a negative potential for UNS N08367, the potential of UNS S31603 was approximately 0.85 V. Thus, it seems that UNS S31603 did not reach the potential of pitting corrosion.

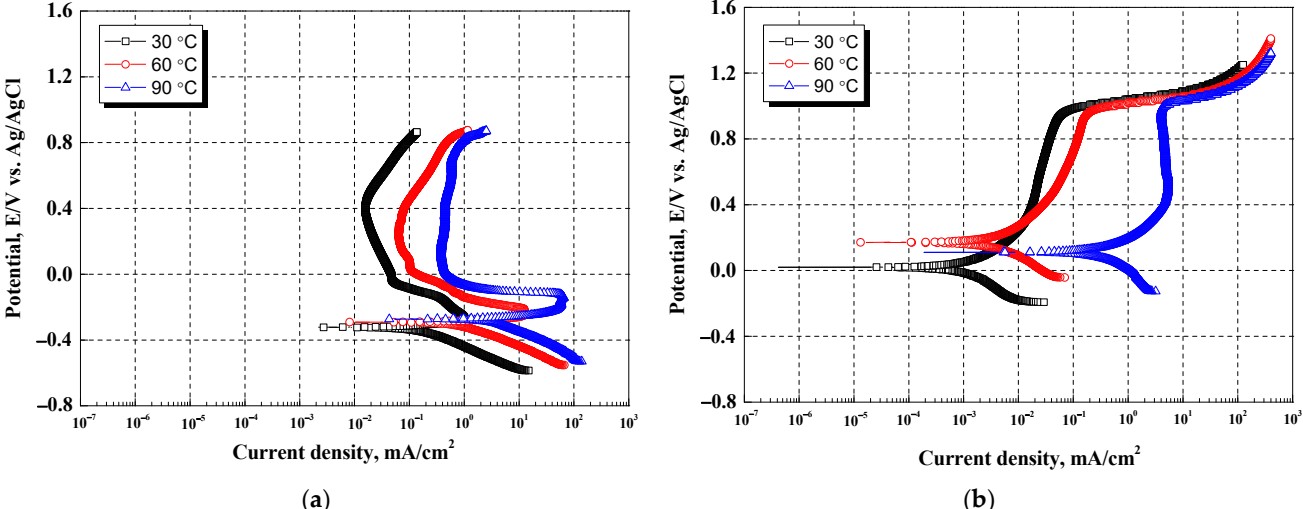

(**a**)     (**b**)

**Figure 1.** Potentiodynamic polarization curves of UNS S31603 and UNS N08367 in a modified green death solution. (**a**) Polarization curves for UNS S31603; (**b**) polarization curves for UNS N08367.

Figure 2 presents the corrosion current density values at different temperatures. In both UNS S31603 and UNS N08367, the corrosion current density increased with temperature. Therefore, it can be assumed that the dissolution reaction became active as the temperature increased. It has been reported that the corrosion current density is low at low temperatures because the surface tension and viscosity of oxide films are high [21]. The corrosion current densities (calculated using the Tafel extrapolation method) at 30 °C, 60 °C, and 90 °C were $1.48 \times 10^{-1}$, $8.12 \times 10^{-1}$, and 2.25 mA/cm$^2$, respectively, for UNS S31603, and $1.15 \times 10^{-3}$, $4.60 \times 10^{-3}$, and $5.01 \times 10^{-1}$ mA/cm$^2$ for UNS N08367. Thus, UNS N08367 exhibited lower corrosion current densities than UNS S31603 at all temperatures and at 90 °C—the most corrosive condition—UNS S31603 had a value approximately 4.5 times higher than UNS N08367. The lower corrosion current densities of UNS N08367 are attributable to the contents of molybdenum (Mo), N, chromium (Cr), and copper (Cu), which affect passive oxide films. The higher the content, the better the stability and formation ability of passive oxide films. The Mo, Cr, N, and Cu contents of UNS N08367 were approximately 3.17, 1.23, 19, and 2 times higher, respectively, than those of UNS S31603.

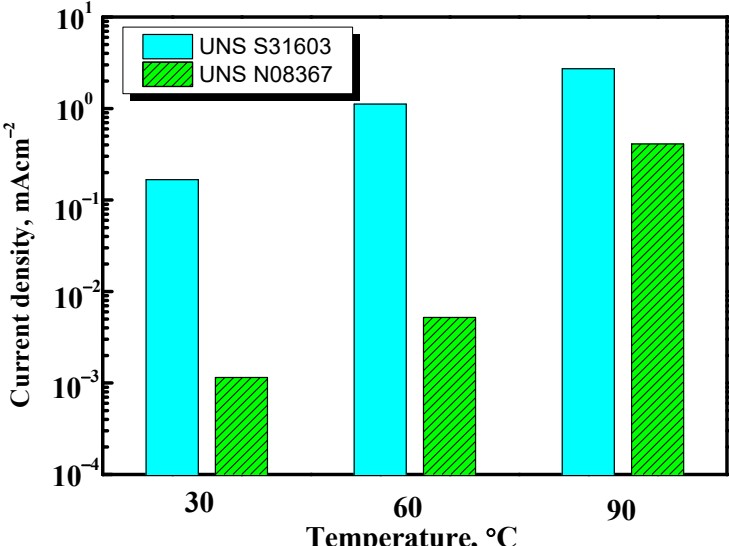

**Figure 2.** Tafel analysis results after the electrochemical experiments on UNS S31603 and UNS N08367 in a modified green death solution.

Figure 3 presents the SEM images and EDS analysis results of UNS S31603 and UNS N08367 after potentiometric polarization experiments in a green death solution at 60 °C. In both specimens, chromium, molybdenum, nickel, copper, iron, and oxygen elements were detected. The component content of each element on the surface of UNS S31603 was detected as Cr = 15 wt%, Mo = 2 wt%, O = 1.5 wt%, Ni = 11 wt%, Cu = 0.8 wt%, and Fe = 69 wt%, while that on the surface of UNS N08367 was detected as Cr = 21 wt%, Mo = 7 wt%, O = 1.9 wt%, Ni = 25 wt%, Cu = 1.5 wt%, and Fe = 44 wt%. Clearly, the contents of chromium, molybdenum, and copper, which are elements affecting corrosion resistance, in UNS N08367 are generally increased compared with those in UNS S31603. In general, a passivation oxide film in an acidic solution reacts with chromium ions ($Cr^{3+}$) and oxygen ions ($O^{2-}$) to form the crystal structure of chromium oxide ($2Cr^{3+} + 3O^{2-} \rightarrow 2Cr_2O_3$) [22]. In this research, it is believed that oxygen (O) was detected because a chromium oxide layer was formed in the two specimens.

Figure 4 presents a schematic diagram of the passive oxide film formation mechanism based on the bipolar model of passive oxide films formed on the surface of austenitic stainless steel [23]. Stainless steel containing Mo indicates greater corrosion resistance due to passivation oxide films formed as bipolar duplex membranes [23]. These membranes are composed of a cation-selective layer and an anion-selective layer, which begin with molybdenum ions ($Mo^{6+}$) and chromium ions ($Cr^{3+}$). As shown in Figure 4a, the stainless steel alloy with $Mo^{6+}$ reacted with $H_2SO_4$ and HCl in the modified green death solution, forming $CrO_4^{2-}$ and $MoO_4^{2-}$ anions on the outer layer of the passive film [23]. These anions exhibited the properties of cation-selective layers, suppressing the invasion of sulfate ions ($SO_4^{2-}$), chloride ions ($Cl^-$), and hydroxyl ions ($OH^-$) contained in the modified green death solution, thereby reducing the stainless steel's oxidation reaction rate. Chromium hydroxide ($Cr(OH)_3$) and $OH^-$ were decomposed by a dehydrogenation reaction, forming oxygen ions ($O^{2-}$) and hydrogen ions ($H^+$).

$$Cr(OH)_3 \xrightarrow{H_2O} Cr^{3+} + 3OH^-, OH^- \rightarrow H^+ + O^{2-} \tag{2}$$

$O^{2-}$ moved to the metal surface through an anion-selective layer (Figure 4b) and combined with Cr on the metal surface, forming a chromium oxide ($Cr_2O_3$) layer, thereby stabilizing the oxide film.

| SEM image | Cr | Mo |
|---|---|---|
| (a) | 15 wt% | 2 wt % |
| **O** | **Ni** | **Cu** |
| 1.5 wt % | 11 wt % | 0.8 wt % |
| SEM image | Cr | Mo |
| (b) | 21 wt % | 7 wt % |
| **O** | **Ni** | **Cu** |
| 1.9 wt % | 24 wt % | 1.5 wt % |

**Figure 3.** EDS analysis of UNS S31603 and UNS N08367 after potentiodynamic polarization experiment in modified green death solution at 60 °C (**a**): UNS S31603 (**b**): UNS N083637.

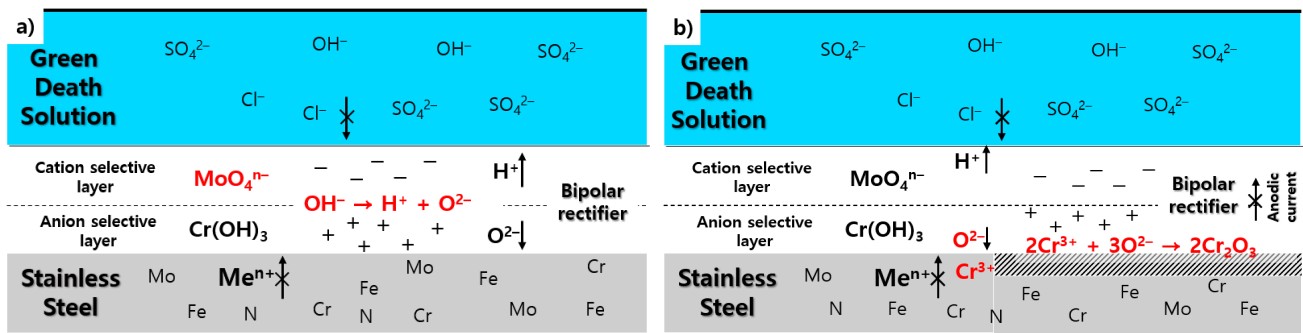

**Figure 4.** Schematic diagrams of the bipolar model and passive film formation mechanism of stainless steel in a modified green death solution. (**a**) Decomposition of chromium hydroxide and hydroxide ions → formation of oxygen ions and hydrogen ions, (**b**) oxygen ions move to the metal surface → formation of chromium oxide layer.

$$2Cr^{3+} + 3O^{2-} \rightarrow 2Cr_2O_3 \tag{3}$$

$H^+$ ions were discharged into the solution through the cation-selective layer, leading to a decrease in acidity [20,23]. N added to stainless steel affects the corrosion current density. It reacts with $OH^-$, generating nitrates ($NO^{3-}$), which form a bipolar duplex membrane on the outer layer of the passive film. Like bipolar duplex membranes formed by Mo, this membrane reduces the corrosion current density [20]. Consequently, Mo, Cr, and N are considered to stabilize the formation of passive film layers. Moreover, the Cr and $Cr_2O_3$ contents in passivation films can be increased using an accelerator that induces the penetration of Cr and it has been reported that corrosion resistance is affected by the Cr and $Cr_2O_3$ contents in the film rather than by the film's thickness [20]. By promoting the reaction of Cr and oxygen on the metal surface through the bipolar duplex membrane formed by Mo and $NO^{3-}$, the $Cr_2O_3$ content in the passivation film increases. Therefore, the corrosion resistance of stainless steel containing Mo is superior to that of stainless steel with conventional oxide films. Furthermore, a higher Cu content decreases the corrosion current density [15].

Figure 5 illustrates the mechanism of the corrosion current density reduction by forming a Cu layer with a higher content than UNS S31036 on the surface of UNS N08367. This process began with the generation of electrons through the oxidation reaction of Fe in a harsh corrosive environment. The generated electrons finally combined with $Cu^{2+}$ ions, and the recrystallized Cu was deposited on the metal surface, forming Cu layers. $Cu^{2+}$ and $Fe^{2+}$ ions were oxidized by dissolved oxygen. However, due to the difference in potential between the two, $Cu^{2+}$ ions were reduced to Cu by receiving electrons generated by $Fe^{2+}$ and were concentrated on the metal surface, as in electroplating (Figure 5a,b). Cu was again ionized into $Cu^{2+}$ by the oxygen reduction reaction (Figure 5c). After the base material was exposed due to the ionization of $Cr^{2+}$, the binding energy of $Cu^{2+}$ was higher than the adsorption energy of $Cl^-$. Thus, $Cu^{2+}$ reacted with the electrons generated by Fe, forming porous CuCl(s) (Figure 5d). Due to its porosity, CuCl(s) exposed the base material to the corrosive environment, and the exposed Fe was oxidized by dissolved oxygen, generating electrons (Figure 5e). CuCl(s) reacted with the generated electrons, generating Cu. Cu was then deposited on the metal surface and the process was repeated (Figure 5f) [24–28].

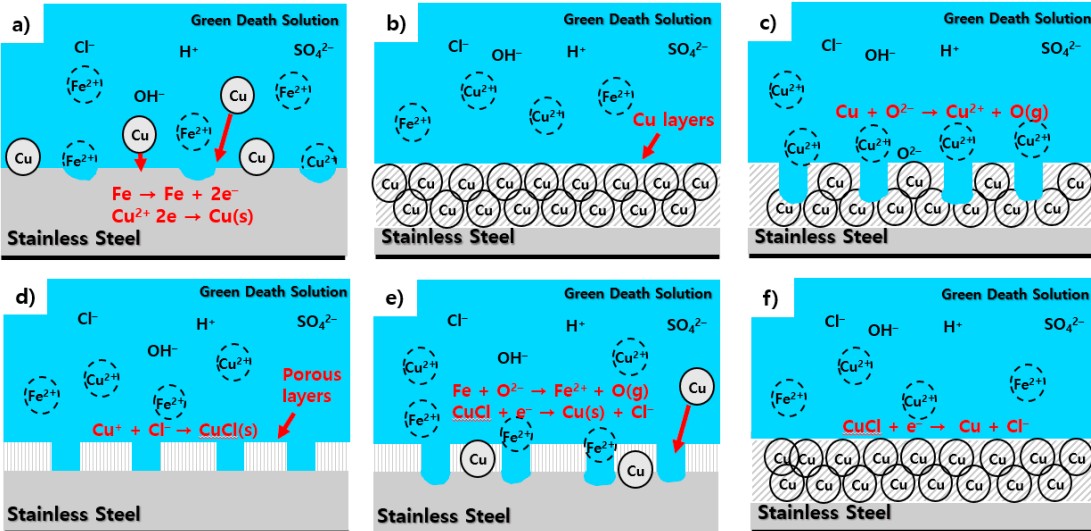

**Figure 5.** Schematic diagrams of the selective dissolution and redeposition mechanisms of stainless steel in a modified green death solution. (**a**) Reduction reaction of copper ion, (**b**) deposition of copper, (**c**) re-oxidation reaction of copper layer, (**d**) formation of copper chloride (porous film), (**e**) re-oxidation of substrate (Fe), (**f**) redeposition of copper (reduction reaction).

Figure 6 presents the corrosion area ratios at different temperatures obtained after the electrochemical experiments. In both UNS N08367 and UNS S31603, the corrosion area ratios tended to increase as the temperature increased. Pitting corrosion caused by HCl

and H$_2$SO$_4$ was observed in the entire specimens. At 30 °C, scratches due to polishing and partially corroded surfaces were equally observable. Under the condition of 60 °C, the corroded surface was more clearly observed than the polishing scratch and it is thought that the influence of crevice corrosion was slightly observed because of the self-manufactured holder. At 90 °C, polishing scratches were not observed; only corrosion was observed in this highly corrosive condition. In the case of UNS N08367, only polishing scratches were observed at 30 °C, while corrosion was observed only locally. It can be assumed that as the Mo, Cr, N, and Cu contents increased, the ability to form passive oxide films improved and stabilized, leading to high corrosion resistance. Polishing scratches and corrosion damage were observed together at 60 °C, but the resistance to crevice corrosion was remarkably high, resulting in uniform damage throughout. Only corrosion damage was observed in the 90 °C temperature condition. The damage rates of UNS S31603 at 60 °C and 90 °C were 5.62% and 8.44%, respectively, while those of UNS N08367 were 3.53% and 5.75%, respectively. The lower damage rates of UNS N08367 indicate that it had higher resistance to pitting corrosion than UNS S31603.

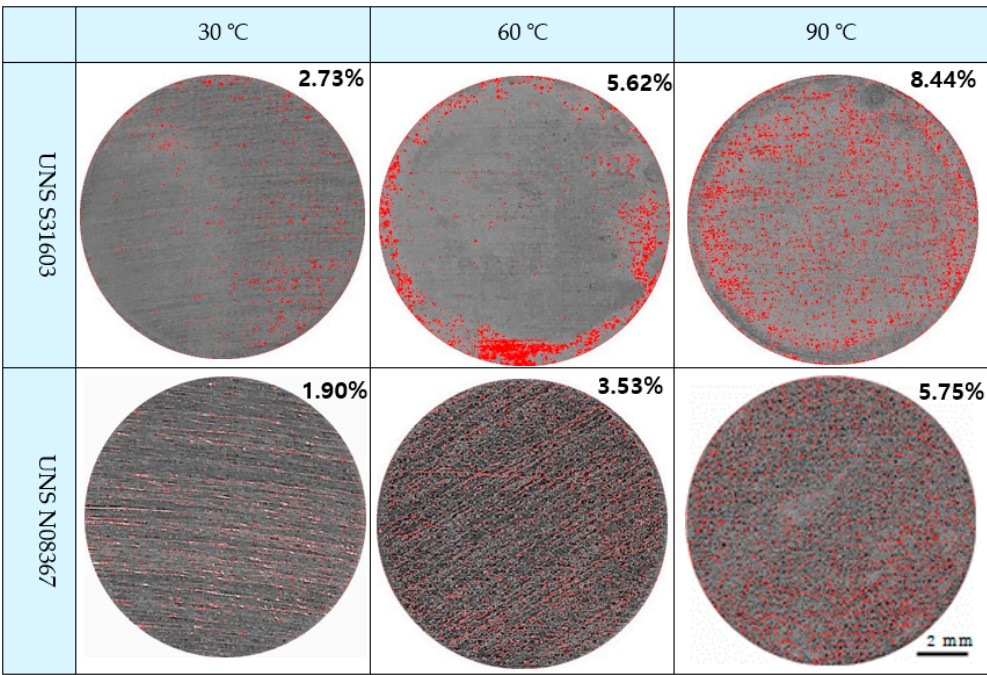

**Figure 6.** Appearance and corrosion area ratios of UNS S31603 and UNS N08367 after the electrochemical experiments in a modified green death solution with temperatures.

Figure 7 depicts surfaces observed using SEM after the electrochemical experiments at different temperatures. At 30 °C, UNS S31603 exhibited local pitting damage due to Cl$^-$ and SO$_4^{2-}$. Pitting damage can mainly be attributed to Cr, Ni, and Mo, which are contained in the two types of stainless steel. Alloys containing Mo, Cr, or Ni are greatly affected by the destruction of films due to halogen ions, including Cl$^-$ [29]. The destruction of passive films by Cl$^-$ and SO$_4^{2-}$ begins locally, in areas where the films are thin and structurally uneven. In this case, the locally damaged passive film acted as an anode, while the periphery acted as a cathode, so local corrosion proceeded rapidly due to the galvanic effect. At 60 °C, corrosion grew along the grain boundary, which was relatively poor, and pitting corrosion developing in the grains was also observed. At 90 °C, intergranular corrosion and pitting corrosion developing in the grains grew further, resulting in extremely severe damage. In UNS N08367, intergranular corrosion was clearly observed. At 30 °C, pitting damage due to local corrosion was observed. At 60 °C, corrosion along the relatively poor grain boundary and shallow pitting corrosion developing in the grains were also

observed. At 90 °C, local corrosion by intergranular corrosion was more clearly observed than local corrosion by intragranular corrosion.

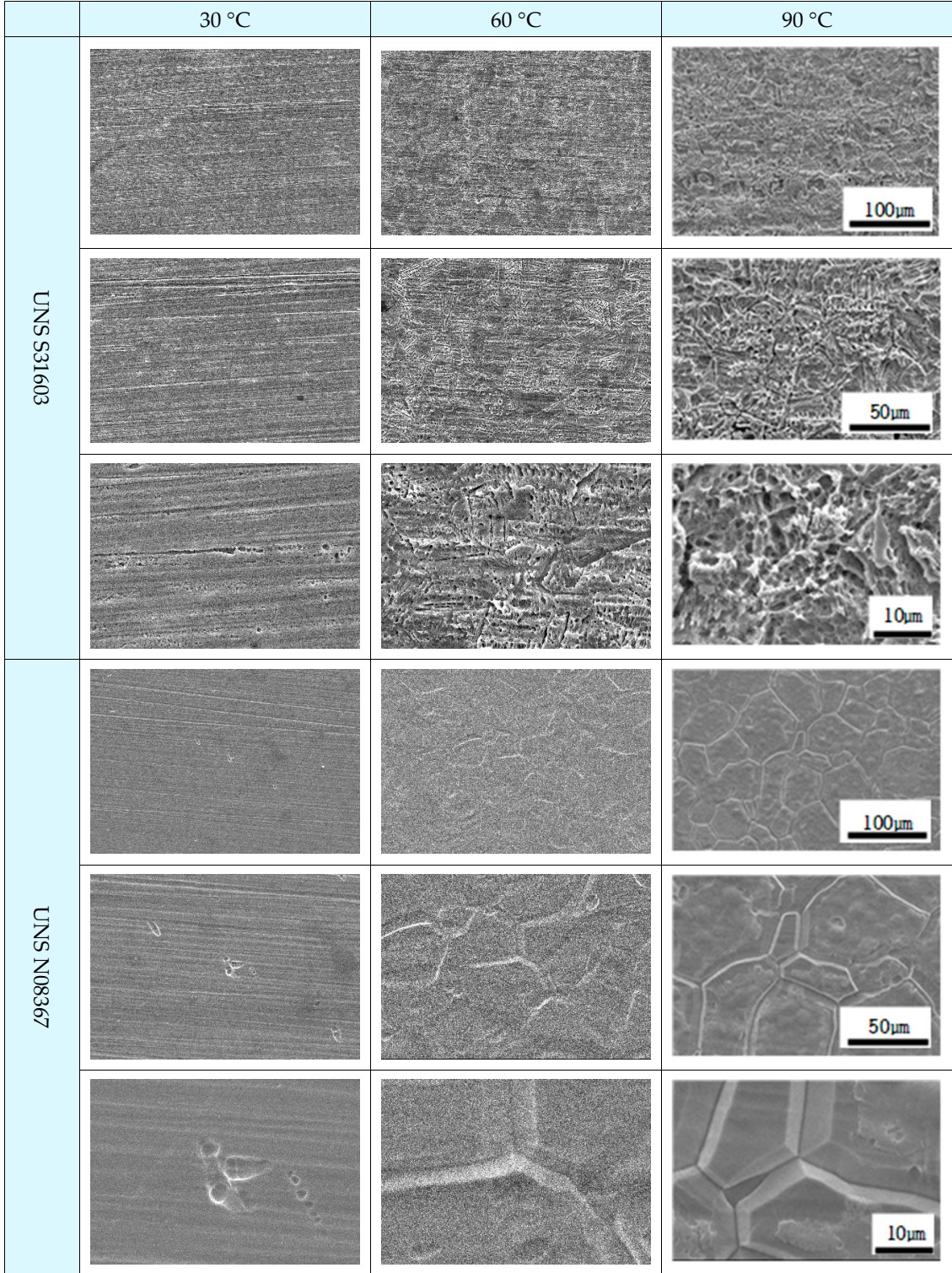

**Figure 7.** Surface morphologies of UNS S31603 and UNS N08367 after the electrochemical experiments in a modified green death solution with temperatures.

Figure 8 presents the results of the 3D microscopic analysis of the surfaces and the maximum damage depths after the electrochemical experiments at different temperatures. Overall, the damage depth increased with temperature due to the high mobility of chloride

ions and sulfate ions [30]. The maximum damage depths of UNS N08367 were generally shallower than those of UNS S31603. The maximum damage depths of UNS S31603 and UNS N08367 were 25.84 and 10.61 µm, respectively, at 30 °C; 49.85 and 44.38 µm, respectively, at 60 °C; 88.34 and 64.32 µm, respectively, at 90 °C. Under the most severe corrosion condition, the maximum damage depth of UNS S31603 was approximately 1.37 times greater than that of UNS N08367. In the case of UNS S31603, the difference in maximum damage depth increased with temperature. The difference in maximum damage depth between 30 °C and 60 °C was 24.01 µm, while the difference between 60 °C and 90 °C was 38.49 µm. Conversely, in the case of UNS N08367, the difference in maximum damage depth decreased with temperature. The difference in maximum damage depth between 30 °C and 60 °C was 33.77 µm, while the difference between 60 °C and 90 °C was 19.94 µm. This is because as the temperature increased, the movements of Cr, Mo, N, O, H, and other electrons became more active. Thus, the oxygen ions in the anion-selective layer of the bipolar duplex membrane quickly moved to the metal surface, so the repassivation of the film progressed rapidly. On the other hand, the $H^+$ ions in the cation-selective layer moved to the solution, leading to decreased activity, which resulted in shallower maximum damage depths.

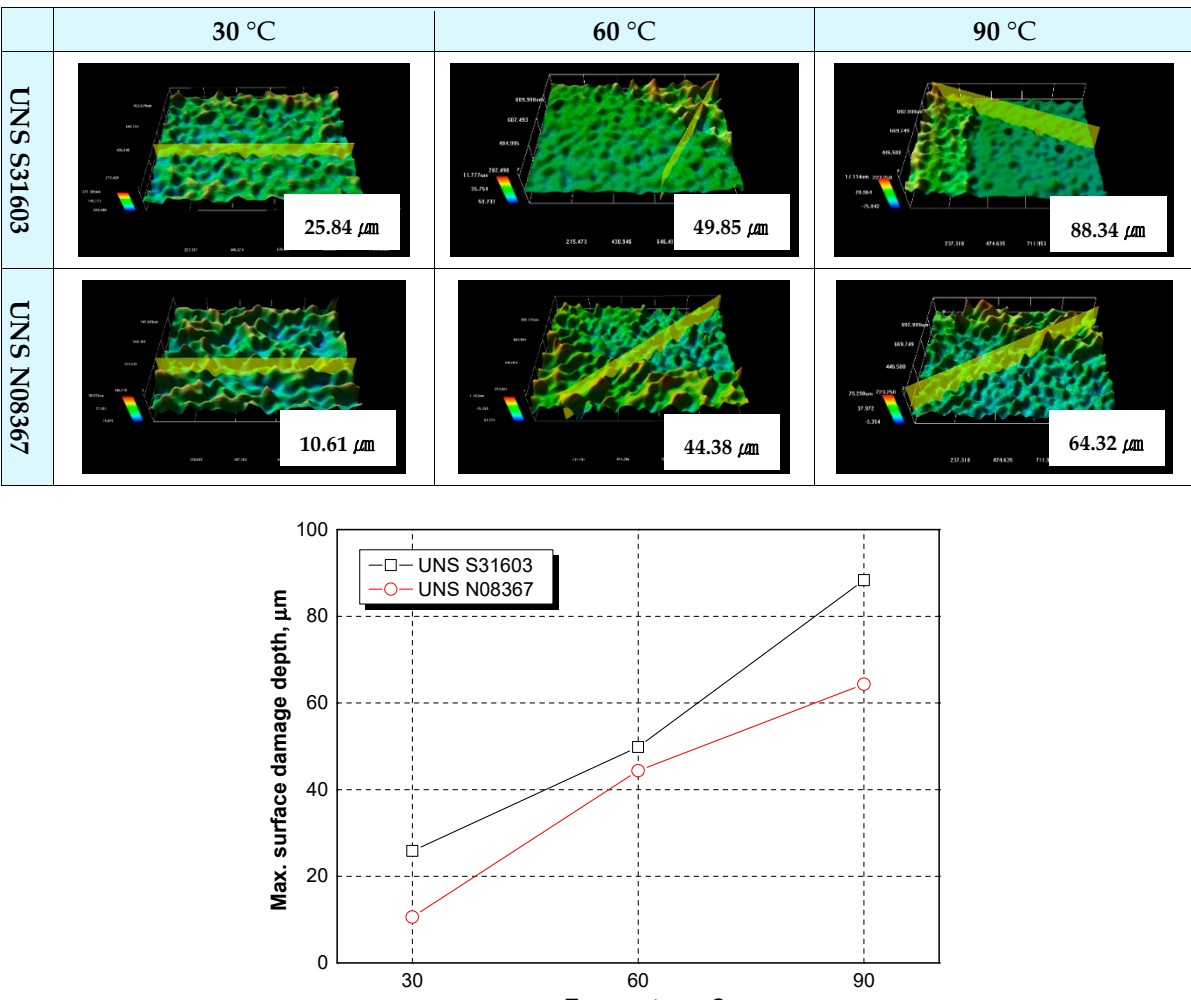

**Figure 8.** Surface damage depths of UNS S31603 and UNS N08367 after the electrochemical experiments in a modified green death solution with temperatures.

Figure 9 shows the corrosion rates calculated according to ASTM-G102 (standard practice for calculation of corrosion rates and related information from electrochemical measurements) after the electrochemical experiments at different temperatures. Overall,

UNS N08367 exhibited lower values than UNS S31603 and its corrosion rate at 90 °C was approximately 4.5 times lower than that of UNS S31603. This indicates that UNS N08367 is superior to UNS S31603 in terms of passive film-forming ability and stability due to Mo, Cr, and N. The corrosion rate differences between UNS S31603 and UNS N08367 increased with temperature. The differences between the two types of stainless steel at 30 °C, 60 °C, and 90 °C were approximately 1.56, 8.6, and 18.7 mm/yr, respectively. It can be assumed that although corrosion increased in both types of stainless steel as the temperature rose because the active dissolution reactions increased, the corrosion rate of UNS N08367 was slower than that of UNS S31603 because of its more stable passive oxide films. Jang et al. compared the corrosion rate by weight loss with the corrosion rate by depth damage after a cavitation erosion experiment and expressed the relationship between the two as follows [31]:

$$\alpha = \frac{\text{Corrosion rate (by damage depth)}}{\text{Corrosion rate (by weight loss)}} = \frac{\text{Localized corrosion}}{\text{General corrosion}} \tag{4}$$

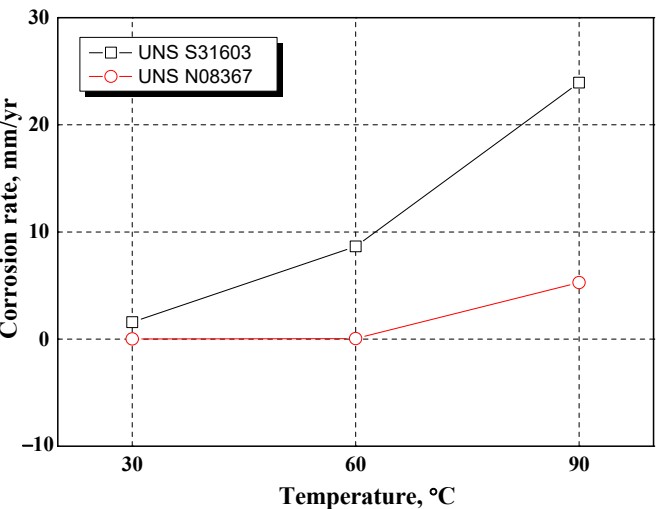

**Figure 9.** Corrosion rates (mm/yr) of UNS S31603 and UNS N08367 after the electrochemical experiments in a modified green death solution at different temperatures.

According to Equation (2), the corrosion rate according to damage depth is $\alpha$ times higher than the corrosion rate according to weight loss. In this investigation, the corrosion rate according to weight loss (general corrosion) was replaced with the corrosion rate based on the corrosion current density to compare the degrees of local corrosion of the specimens.

Table 3 indicates the process of calculating the $\alpha$ values using the corrosion rates (mm/yr) calculated based on the corrosion current densities and maximum damage depths. The obtained $\alpha$ values were divided by 1000. For UNS S31603, the resultant values at 30 °C, 60 °C, and 90 °C were 0.19, 0.06, and 0.04, respectively. This means that the corrosion rates according to maximum damage depth were 0.19, 0.06, and 0.04 times higher, respectively, than those obtained based on the corrosion current density. In the case of UNS N08367, the resultant values at 30 °C, 60 °C, and 90 °C were 10.45, 8.46, and 0.14, respectively.

**Table 3.** Alpha($\alpha$) value calculation process after the electrochemical experiments on UNS S31603 and UNS N08367 in a modified green death solution with temperatures.

|  | E.W (1/NEQ) | D (g/cm$^3$) | Temperature (°C) | Iccor. ($\mu$A/cm$^2$) | MPY | mm/Year |
|---|---|---|---|---|---|---|
|  |  |  | 30 | 148 | 61.96 | 1.57 |
| UNS S31603 | 25.73281 | 7.99 | 60 | 812 | 339.97 | 8.64 |
|  |  |  | 90 | 2250 | 942.033 | 23.93 |

**Table 3.** *Cont.*

| | E.W (1/NEQ) | D (g/cm³) | Temperature (°C) | Iccor. (µA/cm²) | MPY | mm/Year |
|---|---|---|---|---|---|---|
| UNS N08367 | 25.66358 | 8.06 | 30 | 1.15 | 0.47 | 0.012 |
| | | | 60 | 4.61 | 1.90 | 0.048 |
| | | | 90 | 501 | 207.37 | 5.27 |
| | Temperature (°C) | | Corroded depth (µm) | | mm/Year | $\alpha/1000$ |
| UNS S31603 | 30 | | 25.84 | | 305.4 | 0.19452 |
| | 60 | | 49.85 | | 589.1 | 0.06818 |
| | 90 | | 88.34 | | 1044.1 | 0.04363 |
| UNS N08367 | 30 | | 10.61 | | 125.40 | 10.450 |
| | 60 | | 44.38 | | 406.34 | 8.465 |
| | 90 | | 64.32 | | 760.21 | 0.144 |

Figure 10 shows the $\alpha$ values. In general, UNS N08367 had higher values than UNS S31603, indicating higher local damage ratios. As shown in Figure 6, intergranular and pitting corrosion developed simultaneously in UNS S31603, while UNS N08367 exhibited local corrosion due to intergranular corrosion. Moreover, UNS N08367 exhibited a transpassive section (Figure 1b), whereas UNS S31603 showed no transpassive section (Figure 1a). Thus, it can be assumed that the current density of UNS N08367 increased rapidly due to the destruction of its passive state, leading to increased local damage.

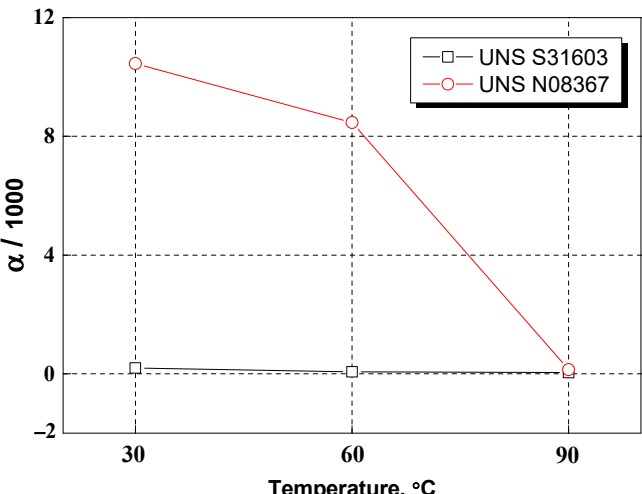

**Figure 10.** Alpha($\alpha$) value UNS S31603 and UNS N08367 after electrochemical experiment in modified green death solution with temperature.

## 4. Conclusions

The following conclusions were obtained following the electrochemical experiments in modified green death solution under temperature conditions.

(1) The Tafel analysis after a potentiodynamic polarization test of UNS N08367 and UNS S31603 in a modified green death solution at various temperatures showed that the corrosion current density of UNS S31603 at 90 °C—the most corrosive condition—was approximately 4.5 times higher than that of UNS N08367, indicating the latter had greater corrosion resistance.

(2) The maximum damage depth (64.3 µm) of UNS N08367 at 90 °C, the most severe corrosion condition, was reduced by about 27% compared with UNS S31603 (88.34 µm).

(3) Local damage increased at higher temperatures due to the greater mobility of chloride ions and sulfate ions, but corrosion resistance increased with higher Mo, Cr, N, and Cu ion contents.

(4) Molybdenum, chromium, and nitrogen form a bipolar membrane passivation film and it is thought that formability and stability are improved as the elemental contents increase. Copper is ionized and redeposited on the metal surface (electroplating effect) to improve corrosion resistance, so this can be interpreted as an improvement.

(5) UNS N08367 had higher α values than UNS S31603, indicating greater local damage. This is because pitting and intergranular corrosion developed simultaneously in the case of UNS S31603, whereas UNS N08367 exhibited local damage due to intergranular corrosion. However, the α values decreased with increasing temperatures, indicating that both local and general corrosion increased.

**Author Contributions:** Conceptualization, S.-J.K. and H.-K.H.; methodology, S.-J.K. and H.-K.H.; validation, S.-J.K. and H.-K.H.; investigation, S.-J.K. and H.-K.H.; resources, H.-K.H.; data curation, S.-J.K.; writing—original draft preparation, H.-K.H.; writing—review and editing, S.-J.K.; visualization, H.-K.H.; supervision, S.-J.K. All authors have read and agreed to the published version of the manuscript.

**Funding:** This research was funded by a research project titled 'Demonstration of aftertreatment systems of Ship's air pollutant ($NO_x/SO_x/PM$) and establishment of their certification system' from the Ministry of Oceans and Fisheries, Korea grant number [20190402] And The APC was funded by a research project titled 'Demonstration of aftertreatment systems of Ship's air pollutant ($NO_x/SO_x/PM$) and establishment of their certification system' from the Ministry of Oceans and Fisheries, Korea.

**Institutional Review Board Statement:** Not applicable.

**Informed Consent Statement:** Not applicable.

**Data Availability Statement:** The data are not publicly available due to the project requirements.

**Conflicts of Interest:** The authors declare no conflict of interest.

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
