# Peer review of "Electrochemical Properties of Super Austenite Stainless Steel with Temperature in a Green Death Solution"

_coatings, doi:10.3390/coatings13010130_

Round 1

Reviewer 1 Report

COMMENTS FOR THE AUTHORS

This work electrochemical experiments on UNS S31603 and UNS N08367 were conducted in a modified green death solution using temperature as a variable, and the 60degrees of damage at temperature were analyzed and compared.

In some cases, the text must be clarified and there are issues that are not accurate. Better and more evidences for some conclusions are required. Without a careful revision this manuscript cannot be accepted.  

1.     In introduction section, more reference should be talked, as well as the research status should be comprehensively.

2.     In Table 2, first line should be <Chemical composition> and <Concentration>, <Sulfuric acid (H2SO4)> and <16.9 vol. %> are in the second line.

3.     Error bar should be added in the Figures.

4.     Evidences should be provided to illustrate the chromium oxide layer,eg. EDS, XPS, XRD results.

5.     Forming rich Cu layers on the surface of stainless steel also need morphologies and microstructure to be proved.

6.     In Fig.5 60℃, why the pitting corrosion in UNS N08367 distribute uniformly, whereas the UNS3163 in the edge?

7.      The resolution is too low in Fig.7.

8.     The conclusion section should be divided several parts, e.g. (1)XXX; (2)XXX; (3)XXX.

Author Response

Please see the attachment(PDF file)
1. Responses

2. Manuscript

Thank you for your review.

Reviewer 2 Report

This paper is quite interesting for dealing with the corrosion problem in the future. The temperature was used as a variable in the analysis of UNS S31603 and UNS N08367 in a modified green death solution, which simulates the environment of a desulfurization device (scrubber). A Tafel analysis revealed that the corrosion current density was approximately 4.5 times greater at the highest temperature (90 °C) than at the lowest temperature (60 °C). SEM surface analysis revealed that pitting and intergranular corrosion occurred concurrently in both samples. The corrosion rates were compared to the damage rates based on maximum damage depth following electrochemical experiments.

This paper cannot be published in its current form. A few points below should be revised by the authors,

1. The title should be corrected and shortened to better describe the contents of the paper.

2. It is necessary to include a one-paragraph explanation of the most recent corrosion treatment in the introduction section so that the significance of this research is clearer. Cite the following related publications in the new paragraph above:

https://www.mdpi.com/2076-3417/10/20/7069 ;

http://electrochemsci.org/papers/vol7/7043274.pdf;

https://www.sciencedirect.com/science/article/pii/S2589014X22000305;

3. Figure 2 and its caption should be on the same page.

4. Figures 3a and 3b should not be split into two pages.

5. In the caption of Figure 3, provide a brief explanation of the differences between Fig. 3a and 3b.

6. Figure 4 caption must include a description of each figure 4a- 4f.

7. equation 4 needs to be tidied up.

8. make table 3 in 1 page, don't separate it into 2 pages.

9. In the conclusion section, the explanation needs to be made more quantitative.

Author Response

(The authors gave the same response as above.)

Round 2

Reviewer 1 Report

Accept

Reviewer 2 Report

Accepted